# Duodenal Gastric Metaplasia and Duodenal Neuroendocrine Neoplasms: More Than a Simple Coincidence?

**DOI:** 10.3390/jcm11092658

**Published:** 2022-05-09

**Authors:** Sara Massironi, Roberta Elisa Rossi, Anna Caterina Milanetto, Valentina Andreasi, Davide Campana, Gennaro Nappo, Stefano Partelli, Camilla Gallo, Miki Scaravaglio, Alessandro Zerbi, Francesco Panzuto, Claudio Pasquali, Massimo Falconi, Pietro Invernizzi

**Affiliations:** 1Division of Gastroenterology, San Gerardo Hospital, University of Milano-Bicocca School of Medicine, 20900 Monza, Italy; c.gallo19@campus.unimib.it (C.G.); m.scaravaglio@campus.unimib.it (M.S.); pietro.invernizzi@unimib.it (P.I.); 2HBP Surgery, Hepatology and Liver Transplantation Unit, ENETS Center of Excellence, Fondazione IRCCS Istituto Nazionale Tumori (INT, National Cancer Institute), 20133 Milan, Italy; 3Pancreatic and Endocrine Digestive Surgical Unit, Department of Surgery, Oncology and Gastroenterology, Università degli Studi di Padova, 35122 Padua, Italy; acmilanetto@unipd.it (A.C.M.); claudio.pasquali@unipd.it (C.P.); 4Pancreatic Surgery Unit, ENETS Center of Excellence, San Raffaele IRCCS, “Vita-Salute” University, 20132 Milan, Italy; andreasi.valentina@hsr.it (V.A.); partelli.stefano@hsr.it (S.P.); falconi.massimo@hsr.it (M.F.); 5ENETS Center of Excellence, Department of Experimental, Diagnostic and Specialty Medicine, Bologna University, St. Orsola-Malpighi University Hospital, 40138 Bologna, Italy; davide.campana@unibo.it; 6ENETS Center of Excellence, Department of Biomedical Sciences, Humanitas University, 20090 Pieve Emanuele, Italy; gennaro.nappo@humanitas.it (G.N.); alessandro.zerbi@hunimed.eu (A.Z.); 7Digestive Disease Unit, ENETS Center of Excellence, Sant’Andrea University Hospital, 00189 Rome, Italy; francesco.panzuto@gmail.com; 8Department of Medical-Surgical Sciences and Translational Medicine, Sapienza University of Rome, 00189 Rome, Italy

**Keywords:** duodenal neuroendocrine neoplasms, duodenal gastric metaplasia, risk factor, epidemiology

## Abstract

Background: Duodenal gastric metaplasia (DGM) is considered a precancerous lesion. No data are available regarding its possible role as a risk factor for duodenal neuroendocrine neoplasms (dNENs). Aims: To assess the prevalence of DGM in a cohort of dNENs. Methods: Subgroup analysis of a retrospective study including dNEN patients who underwent surgical resection between 2000 and 2019 and were observed at eight Italian tertiary referral centers. Results: 109 dNEN patients were evaluated. Signs of DGM associated with the presence of dNEN were reported in 14 patients (12.8%). Among these patients, nine (64.4%) had a dNEN of the superior part of the duodenum, one (7.1%) a periampullary lesion, three (21.4%) a dNEN located in the second portion of the duodenum, with a different localization distribution compared to patients without DGM (*p* = 0.0332). Ten were G1, three G2, and in one patient the Ki67 was not available. In the group with DGM, six patients (35.7%) were classified at stage I, five (28.6%) at stage II, three (21.4%) at stage III, and no one at stage IV. In the group without DGM, 20 patients (31%) were at stage I, 15 (15%) at stage II, 42 (44%) at stage III, and 19 (20%) at stage IV (*p* = 0.0236). At the end of the study, three patients died because of disease progression. Conclusions: our findings might suggest that DGM could represent a feature associated with the occurrence of dNEN, especially for forms of the superior part of the duodenum, which should be kept in mind in the endoscopic follow up of patients with DGM. Interestingly, dNEN inside DGM showed a more favorable staging, with no patients in stage IV. The actual relationship and the clinical relevance of this possible association require further clarification.

## 1. Introduction

Duodenal neuroendocrine neoplasms (dNENs) are rare and heterogeneous tumors that represent up to 3% of all duodenal neoplasms [1]. They usually present in the 6th decade of age with a slight male predominance [2]. Duodenal NENs are usually well-differentiated neoplasms; however, they can be metastatic in up to 55% of cases [3]. Their natural history, clinical characteristics, biological mechanisms, medical or surgical treatment, and prognosis are still poorly understood.

Duodenal NENs originate from aberrant neuroendocrine duodenal cell proliferation; in this microenvironment, complex interactions take place. The recognition of the molecular mechanisms participating in neoplastic transformation could increase the challenging management of this disease. However, at present, little is known about the risk factors of these neoplasms.

The normal mucosa of the duodenum is composed of absorbing columnar enterocytes and secreting goblet cells. Duodenal gastric metaplasia (DGM) is characterized by the replacement of the normal duodenal epithelial cells with gastric mucus-secreting cells that resemble gastric foveolar epithelium. It is commonly considered a precancerous lesion often associated with chronic inflammation. It is generally the consequence of abnormally high production of gastric acid triggered by *Helicobacter Pylori* (HP) infection [4]. When hypersecretion reaches the duodenum, the enterocytes of the villi react with apical mucin metaplasia to mitigate the unwanted low pH of the microenvironment. Besides HP infection, DGM has been reported in association with other conditions, such as medications (i.e., nonsteroidal anti-inflammatory drugs; NSAIDs), celiac disease [5], and Crohn’s disease involving the duodenum [6]. However, DGM has been described also in the absence of all these conditions, although its actual etiology in the latter group of patients is unclear. Furthermore, DGM usually disappears following HP eradication [7], whereas the natural course of DGM in celiac patients or patients without a recognized cause, even with the application of a strict gluten-free diet, is still poorly known.

It still remains a question of debate whether DGM could represent a neoplastic risk factor. A high frequency (40.5%) of DGM has been found in duodenal adenomas [8]. It might be possible that metaplasia precedes the neoplastic transformation as has been reported in other gastrointestinal malignancies including esophagus (intestinal metaplasia in Barrett’s esophagus–dysplasia–carcinoma sequence [9] and stomach [10] and colorectal cancer [11]). Furthermore, DGM has been associated with genetic alterations, such as GNAS and KRAS mutations, which are involved in different types of tumors including duodenal adenocarcinoma.

However, no data are available regarding the possible role of DGM as a risk factor for the occurrence of dNEN. Taking into account these observations and the lack of clear-cut data regarding the natural history of dNEN, we aimed at assessing the prevalence of DGM in a cohort of dNENs. The secondary aim was to explore whether the presence of DGM had any impact on the characteristics or outcome of the current cohort of dNENs.

## 2. Materials and Methods

We performed a subgroup analysis of a retrospective study [3] including all consecutive patients with dNEN, who underwent surgical resection between 2000 and 2019 and who were observed at eight Italian tertiary referral centers.

All data were retrieved at the center where each patient had been diagnosed and followed up. Participating study centers sent the anonymized data of patients to the lead center. The study’s inclusion criteria were age > 18 years, histological diagnosis of dNEN of any grade and stage, surgical treatment of the primary tumor, availability of complete histopathological examination of the surgical specimen, and clinical data with a minimum 3 month follow up after diagnosis. The exclusion criteria were histological findings of mixed neuroendocrine non-neuroendocrine neoplasms (MiNEN), age < 18 years, the use of experimental drugs during the 2 months preceding inclusion in this study, and pregnancy or breastfeeding status. Due to the retrospective nature of this study, ethical approval was waived.

The tumor characteristics analyzed comprised the site and the size of the primary tumor, number of lesions, grade, and stage (i.e., localized, regional, distant, and unknown). The patient’s characteristics included the age at first diagnosis, the presence of genetic syndrome (i.e., multiple endocrine neoplasia (MEN)-1), and the presence of functioning neoplasms.

Medical history data were collected and recorded by physicians in electronic health records, comprising the clinical history, age at diagnosis, treatments received, clinical and biochemical parameters, radiological imaging, endoscopy examinations, and nuclear medicine imaging were recorded and evaluated at each referral center. The type of surgical intervention was recorded for all the patients.

Neoplasms were classified according to the WHO 2019 classification [12] and staged according to the current European Neuroendocrine Tumor Society (ENETS) TNM clinical staging [13].

For each included patient, the endoscopic or surgical specimen and related histopathological data were assessed to verify the presence or absence of DGM. Concomitant treatment with proton pump inhibitors (PPIs) was recorded.

### Statistical Analysis

Descriptive statistics were used to summarize the data. Continuous variables with normal distribution were expressed as the median (i.e., range); categorical variables were reported as the count (i.e., percentage). All data were tested for distribution normality by the Kolmogorov–Smirnoff test. The differences between groups were assessed with the Mann–Whitney test and the Kruskal–Wallis test as appropriate. Comparisons between groups were assessed using the χ^2^ test or Fisher’s exact test. The analyses were carried out using GraphPad Prism version 6.00 (GraphPad Software, San Diego, CA, USA).

## 3. Results

From 2000 to 2019, 109 patients with histologically confirmed dNEN were included in the study as previously reported (Figure 1).

The DGM associated with a dNEN was reported in 14 patients (12.8%). None of these patients had a concomitant HP infection, celiac disease, or Crohn’s disease. Concomitant use of NSAIDs was excluded for all 14 patients.

The baseline characteristics of these 14 patients were compared to the clinical features of the remaining 95 patients without signs of DGM (Table 1). We observed a male prevalence in both groups, whereas the patients with DGM were older (61.5 versus 58 years old), even if this difference was not statistically significant. In the two groups, the median diameter of the neoplasms was similar (being quite small, namely, 15 in patients without DGM and 11 mm in patients with DGM), and the majority of tumors were single. Location of the primary NEN was significantly different between the two groups (*p* = 0.0332): among the 14 patients with DGM, 9 had a dNEN of the superior part of the duodenum (64.4%), 1 had a periampullary neoplasm (7.1%), in 3, the dNEN was located in the second portion of the duodenum (21.4%), whereas in 1 patient the location was not specified. Among the 95 patients with dNEN without DGM, the majority (42.1%) showed periampullary tumors.

As concerning grading, among the patients with DGM, 10 were G1; 3 G2; while in 1 patient the ki67 was not specified. None of the tumors inside DGM was a poorly differentiated neoplasm. Among the 95 patients without DGM, 56 were G1; 23 G2; 7 G3; whereas in 9 patients the Ki67 was not available.

The staging had a significantly different distribution between the two groups (*p* = 0.0236); in the group with DGM, six patients were classified as stage I; five as stage II; three as stage III; no one at stage IV. In the other group without DGM, 20 patients were at stage I; 15 at stage II; 42 at stage III; 19 at stage IV.

The type of surgery was significantly different between the two groups (*p* = 0.0007): 3 out of the 14 patients (21.5%) with DGM underwent pancreaticoduodenectomy, 6 (42.8%) duodenotomy with enucleation, and 5 (35.7%) partial duodenectomy and lymphadenectomy. Among the 95 patients without DGM, 58 (61%) underwent pancreaticoduodenectomy, 4 (4.2%) total pancreatectomy, 28 (29.5%) duodenotomy and enucleation, and five (5.3%) partial duodenectomy and lymphadenectomy.

In the group of 14 patients with DGM, the 5 patients at stage III presented with lymph node metastases at diagnosis and received treatment with somatostatin analogs (SSAs), which were continued after surgery.

One patient out of 14 (7.1%) with DGM-associated dNEN and 17 out of 95 (17.9%) with dNEN not associated with DGM were diagnosed with MEN-1 syndrome, without any significant difference in the percentage of MEN-1. In both groups, the majority of the tumors were nonfunctioning. Five patients (35.7%) were treated with proton pump inhibitors (PPIs) versus 31 patients in the group without DGM (32.6%).

At the end of the study, three patients out of the 14 with DGM (21.4%) were dead, of which only one was due to the fact of disease progression (occurrence of distant liver metastases treated with SSA and chemotherapy). In the group without DGM, 18 patients passed away (18.9%), 13 due to the fact of disease progression.

## 4. Discussion

Duodenal NENs are rare neoplastic lesions born by the aberrant proliferation of the neuroendocrine epithelial cells of the duodenal mucosa [3]. To date, no specific risk factors for the development of dNEN are known; thus, more efforts should be made to identify patients at risk (i.e., by the identification of preneoplastic lesions) in order to develop disease-specific surveillance [14]. In our multicenter study, we demonstrated that the existence of a DGM characterized a non-negligible percentage of dNEN cases, suggesting this could represent a potential risk factor for dNEN. DGM was, in fact, found in almost 13% of the entire cohort of 109 dNEN patients surgically treated.

However, the actual percentage of DGM in the general population is poorly known as variable percentages have been reported in the literature [15,16], and this might be worthy of investigation.

The percentage reported in the current paper was, conversely, quite far from the high percentage described for duodenal adenomas in which DGM has been found to be as high as 40.5%, even if this percentage could be underestimated, considering this alteration has never been described in relation with dNENs; therefore, one can hypothesize that with increasing awareness, this finding could have a greater frequency.

Many studies have demonstrated that several lesions that were thought to be metaplastic may actually represent a potential precursor of common neoplasms. For example, colorectal hyperplastic polyps, which exhibit preserved overall crypt organization and no epithelial dysplasia [17], are commonly considered potential precursors of colorectal cancer [18]; similarly, pancreatic intraepithelial neoplasia 1A, which has also been previously regarded as mucinous metaplasia, is now well known to be the earliest stage precursor of invasive pancreatic adenocarcinoma [19]. Likewise, some duodenal tumors, particularly those with a gastric epithelial phenotype, were interestingly proven to arise from DGM [20,21]. DGM is a condition characterized by the metaplastic replacement of the normal duodenal enterocytes by mucinous PAS-positive cells, migrating from the Brunner’s gland ducts and resembling the superficial gastric foveolar epithelium [22]. To be accurate, DGM should be distinguished from duodenal gastric heterotopia (DGH), which is instead characterized by the presence of both the gastric foveolar epithelium and the oxyntic glands. Because of its fully organized structure, DGH has been interpreted as a congenital lesion [23], while DGM is generally regarded as an acquired reactive process caused by chronic inflammatory conditions [24]. The prevalence of DGM is, in fact, higher in patients with HP infection, as it induces a high level of acid burden in the duodenum by increasing gastrin secretion; moreover, the presence of DGM may create a suitable environment for HP colonization, which may exert a cytotoxic effect on mucosal cells and, thus, to the development of further DGM [24,25]. In our study, none of the patients had a concomitant HP infection. As concerned PPIs, five patients in our cohort with DGM were taking PPIs, a fraction not different from the group without DGM, without therefore suggesting a particular etiopathogenetic role of PPIs in the genesis of DGM-related dNEN. However, even if this percentage was not different between the groups, it was surely of relevance in both groups; therefore, one could also hypothesize that PPIs could have a role in the development of duodenal NENs. Unfortunately, this study did not have the power to investigate this topic.

Concerning the possible different characteristics or outcomes of the dNENs arising in DGM, when comparing the two groups, with and without DGM, we observed that the 14 patients with DGM were younger, and most of the dNENs with GDM were located in the superior part of the duodenum. The reason for this is unknown. It could be hypothesized that there are some different etiopathogenetic factors in the genesis of dNEN originating from the first duodenal portion (for example, the effect of hydrochloric acid or different distributions of neuroendocrine cells types, i.e., somatostatin-, gastrin-, serotonin-producing cells). Unfortunately, these are only speculative hypotheses, and this type of study cannot answer this question. Moreover, interestingly, among the 14 patients with DGM, none showed a metastatic disease (none at stage IV) or G3 neoplasms. This might suggest that dNEN associated with DGM could be more similar to the gastric neuroendocrine neoplasms, such as those arising from gastric metaplasia and, therefore, more indolent and lower grade.

Genetic mutations have been also demonstrated to play a potential role in the development of DGM; GNAS and KRAS mutations, for instance, which are generally frequently present in benign/low-grade tumors of the digestive tract [18,26,27,28], were reported to be prevalent in DGM lesions, suggesting that these genetic alterations induce the proliferation of metaplastic epithelium [29]. Given these demonstrations and based on the association observed in our study, one might speculate that the occurrence of DGM is an epiphenomenon of genetic mutations and a chronic inflamed microenvironment [22] together with the gastrin-mediated dysregulation of molecular pathways [4,25,26,27], promoting tumorigenesis, including dNEN formation [28], with possible implications for the endoscopic follow-up of patients with DGM. In the presence of DGM at histology, in fact, it might be possible to consider a closer endoscopic follow up in order to detect early the presence of dNEN.

We acknowledge two main limitations of our study. First, the retrospective nature of the study and the small sample of patients limit the strength of our conclusions; however, dNEN is a rare disease; thus, large prospective cohort studies are difficult. Second, the histological revision of the pathologic samples was not centralized. However, only pathological examinations performed at referral centers for NENs were included in the study, whereas patients with incomplete information were excluded from the analysis.

## 5. Conclusions

In summary, DGM was found in almost 13% of the entire cohort of 109 dNEN patients surgically treated, thus representing a remarkable percentage. Given these data, one might speculate that the presence of DGM could precede the development of dNEN; the common finding of this lesion in the general population as well as the current lack of disease-specific literature allow for the clinical relevance of this possible association to be clarified; however, it should be kept in mind in the endoscopic follow up of patients with DGM that even the lack of clear-cut evidence does not allow to suggest a specific timeline for endoscopic follow up. Moreover, the DGM-related dNEN could have a different natural history compared to the dNEN not related to DGM and, therefore, be susceptible to different treatments. In conclusion, our observations highlight the need for further studies, ideally creating international disease registries, to better understand the biology and natural history of dNEN and, thus, to improve the management of this heterogeneous disease.

## Figures and Tables

**Figure 1 jcm-11-02658-f001:**
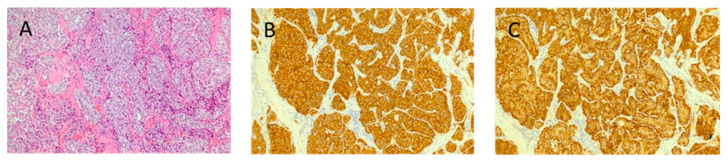
Representative hematoxylin and eosin stain (**A**), synaptophysin (**B**) and chromogranin (**C**) of a duodenal NEN in a 75 year old male patient. The neoplasia was characterized by low mitotic activity (MIB1-labeling index: 0.2%, mitotic index: 0), and a final diagnosis of G1 neuroendocrine tumor was reached (original magnifications: 40×).

**Table 1 jcm-11-02658-t001:** Baseline characteristics of patients with duodenal gastric metaplasia (DGM) associated with duodenal neuroendocrine neoplasms (dNENs) compared to dNEN patients without DGM.

Characteristics	dNENs		*p*
	*w*/*o* DGM n (%)	with DGM n (%)	
Number of patients	95 (87)	14 (13)	
Age (years), median (range)	58 (17–83)	61.5 (32–74)	n.s.
Gender (M/F)	57/38	(11/3)	n.s.
Location Superior part of the duodenum Periampullary Descending duodenum NA	27 (28.4 40 (42.1) 21 (22.1) 7 (7.4)	9 (64.4) 1 (7.1) 3 (21.4) 1 (7.1)	0.0332
Grading (12) G1 G2 G3 NA	56 (58.9) 23 (24.3) 7 (7.3) 9 (9.5)	10 (71.5) 3 (21.4) 0 1 (7.1)	n.s.
Diameter (mm), median (range)	15 (1.5–130)	11 (3–37)	n.s.
Functioning (gastrinoma/somatostatinoma) Nonfunctioning	28 (29.4) (23/4) 69 (70.6)	5 (35.7) (4/1) 9 (64.3)	n.s.
Single Multiple	82 (86.3) 13 (13.7)	11 (78.6) 3 (21.4)	n.s.
Stage (13) I II III IV	20 (21) 15 (15) 42 (44) 19 (20)	6 (42.8) 5 (35.7) 3(21.4) 0	0.0236
Type of surgery Pancreaticoduodenectomy Total pancreatectomy Duodenotomy + enucleation Partial duodenectomy + lymphadenectomy	58 (61) 4 (4.2) 28 (29.5) 5 (5.3)	3 (21.4) 0 6 (42.9) 5 (35.7)	0.0007
MEN-1	17 (17.9)	1 (7.1)	n.s.
Proton pump inhibitor	31 (32.6)	5 (35.7)	n.s.

## Data Availability

Not applicable.

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
