# Peer review of "Duodenal Gastric Metaplasia and Duodenal Neuroendocrine Neoplasms: More Than a Simple Coincidence?"

_jcm, 2022, doi:10.3390/jcm11092658_

Round 1

Reviewer 1 Report

General considerations

Massironi and colleagues provide an elegant and well-written study on a relevant and current topic. The authors bring a compelling insight into the possibility of the relationship between duodenal gastric metaplasia (DGM) and duodenal neuroendocrine neoplasms (dNENs). Indeed, this is a possibility, as epidemiological data are still very scarce to support or not this association. Furthermore, the phylogenetic and histogenetic basis between GDM and dNENs has not yet been clearly and convincingly established. Even so, the present study opening a line of investigation that, in my opinion, deserves to be pursued. I remember that the term duodenal bulb is a radiological term, because the duodenal bulb does not exist as an anatomical structure. I suggest replacing the term “duodenal bulb” with “superior part of the duodenum”.

Specific considerations

I have a couple of remarks to make about this manuscript.

Could the use of proton pump inhibitor among patients with dNENs with or without GDM constitute a bias for the results of this study?

How do the authors explain the significant difference between the number of patients with dNENs without GDM in relation to the number of patients with dNENs and GDM located in the superior part of the duodenum? Does this difference not indicate an absence of association between these two conditions?

Author Response

Reviewer 1

Massironi and colleagues provide an elegant and well-written study on a relevant and current topic. The authors bring a compelling insight into the possibility of the relationship between duodenal gastric metaplasia (DGM) and duodenal neuroendocrine neoplasms (dNENs). Indeed, this is a possibility, as epidemiological data are still very scarce to support or not this association. Furthermore, the phylogenetic and histogenetic basis between GDM and dNENs has not yet been clearly and convincingly established. Even so, the present study opening a line of investigation that, in my opinion, deserves to be pursued.

We thank the reviewer for his positive comment.

I remember that the term duodenal bulb is a radiological term, because the duodenal bulb does not exist as an anatomical structure. I suggest replacing the term “duodenal bulb” with “superior part of the duodenum”.

As suggested by the reviewer, the term “duodenal bulb” has been changed to the term “superior part of the duodenum”.

Specific considerations

I have a couple of remarks to make about this manuscript.

Could the use of proton pump inhibitor among patients with dNENs with or without GDM constitute a bias for the results of this study?

This observation is correct. Concomitant treatment with proton pump inhibitors (PPIs) was recorded for every patient. Five patients (35.7%) were treated with proton pump inhibitors (PPIs) in the group with DGM versus 31 patients in the group without DGM (32.6%), without any difference. Therefore, analyzing the difference between these two groups, this element should not be a bias. However, even if this percentage was not different between the groups, it was surely of relevance in both groups, so one could also hypothesize that PPI could have a role in the development of duodenal NENs. Unfortunately, this study has not the power to investigate this topic, but it should be a matter of debate in future studies. A comment was added in the “Discussion” section (page 7, lines 230-233)

How do the authors explain the significant difference between the number of patients with dNENs without GDM in relation to the number of patients with dNENs and GDM located in the superior part of the duodenum? Does this difference not indicate an absence of association between these two conditions?

We appreciate the reviewer’s observation. Actually, most of the dNENs with GDM were located in the superior part of the duodenum. The reason behind this, is unknown. It could be hypothesized that there are some different etiopathogenetic factors in the genesis of dNEN originating from the first duodenal portion (for example effect of hydrochloric acid, or different distribution of neuroendocrine cell types, i.e. somatostatin-, gastrin-, serotonin-producing cells). Unfortunately, these are only speculative hypotheses, and this type of study cannot respond to this question. A comment was added in the “Discussion” section (page 7, lines 236-241)

Reviewer 2 Report

In the manuscript by Massironi et al., the authors have summarized the clinical observations from 109 patients who developed dNEN. Table 1 in the manuscript illustrates the entire conclusions of the manuscript. The 2 figures reiterate the same conclusion from table 1, which make them redundant to the manuscript. 

Histology data would be more beneficial to illustrate the staging of the disease.

Line 55 needs to be re-phrased

Line 58 needs to be re-phrased

In Figure 1, there is no title for Y axis. 

Author Response

Reviewer 2

In the manuscript by Massironi et al., the authors have summarized the clinical observations from 109 patients who developed dNEN. Table 1 in the manuscript illustrates the entire conclusions of the manuscript. The 2 figures reiterate the same conclusion from table 1, which make them redundant to the manuscript. 

This observation is correct and Figures 1 and 2 have been eliminated.

Histology data would be more beneficial to illustrate the staging of the disease.

A new Figure 1 has been added, reporting the histological features of a duodenal NEN.

Line 55 needs to be re-phrased

We have revised the text to address your concerns and hope that it is now clearer (lines 54-56 in the new version of the manuscript).

Line 58 needs to be re-phrased

We thank the reviewer for pointing this out. We have revised and re-phrased, as suggested (lines 58-59 in the new version of the manuscript).

In Figure 1, there is no title for Y axis. 

Figure 1 has been eliminated, as previously suggested, and replaced by a new Figure reporting the histological features of a duodenal NEN.

Round 2

Reviewer 2 Report

The authors have addressed all the necessary comments.